# 200 GHz single chip microsystems for dynamic nuclear polarization enhanced NMR spectroscopy

Nergiz Sahin Solmaz [1] ✉, Reza Farsi [1] & Giovanni Boero [1]

Dynamic nuclear polarization (DNP) is one of the most powerful and versatile hyperpolarization methods to enhance nuclear magnetic resonance (NMR) signals. A major drawback of DNP is the cost and complexity of the required microwave hardware, especially at high magnetic fields and low temperatures. To overcome this drawback and with the focus on the study of nanoliter and subnanoliter samples, this work demonstrates 200 GHz single chip DNP microsystems where the microwave excitation/detection are performed locally on chip without the need of external microwave generators and transmission lines. The single chip integrated microsystems consist of a single or an array of microwave oscillators operating at about 200 GHz for ESR excitation/detection and an RF receiver operating at about 300 MHz for NMR detection. This work demonstrates the possibility of using the single chip approach for the realization of probes for DNP studies at high frequency, high field, and low temperature.

Nuclear magnetic resonance (NMR) methods are powerful tools employed in many research fields, including physics, chemistry, material science, biology, and medicine. Their importance for industrial and clinical applications is also widely recognized. The use of NMR methodologies for an even wider range of studies is often hindered by the low signal-to-noise ratio (SNR) achievable in the available experimental time. An approach to increase the SNR, applicable to samples of any volume, is to increase the nuclear spin polarization (hyperpolarization) by microwave, optical, or chemistry-based methodologies[1-12]. Dynamic nuclear polarization (DNP) is one of the most powerful and versatile methods to improve the SNR in NMR experiments[1,13-15]. Microwave-driven DNP methods employ a microwave magnetic field that excites the unpaired electrons (added or, rarely, naturally present) in the sample under investigation into electron spin resonance (ESR). The ESR excitation of the unpaired electrons increases the polarization of the nearby nuclear spins well above the thermal equilibrium value, producing an effective increase in the SNR up to 660 for $^1$H nuclei (and more for nuclei with a lower gyromagnetic ratio).

The main difficulty in performing NMR experiments in microscopic samples is the poor SNR. In some experiments, increasing the

averaging time to improve the SNR is not possible, such as in the case of biological samples which might evolve/deteriorate over the required experimental time to achieve a sufficiently large SNR. In these cases, even a modest DNP enhancement would be of crucial importance for the feasibility of the experiment. The single-chip integration of the sensitivity-relevant part of NMR, ESR, and DNP-enhanced NMR detectors is a promising approach to improve the limit of detection, especially for nanoliter and sub nanoliter samples. During the last two decades, the separate integration on a single chip of the front-end electronics of inductive NMR spectrometers[16-30] as well as ESR spectrometers[31-39] have been demonstrated. Recently, the co-integration on a single silicon chip of the front-end electronics of an NMR and an ESR detector has been demonstrated[40]. This combination of sensors allowed us to perform DNP experiments at 10.7 GHz (ESR)/16 MHz (NMR) using a single chip integrated microsystem of about 2 mm². Overhauser $^1$H DNP experiments in liquid samples at room temperature with NMR enhancements as large as 50 have been reported.

DNP-enhanced NMR experiments are performed in a wide range of magnetic fields, from zero to more than 20 T[1,11,41-44]. Operation at higher magnetic fields allows for larger chemical shifts and potentially

$^1$Institute of Electrical and Micro Engineering (IEM) and Center for Quantum Science and Engineering (QSE) École Polytechnique Fédérale de Lausanne (EPFL), 1015 Lausanne, Switzerland. ✉e-mail: nergiz.sahin@epfl.ch

larger SNR. The larger chemical shifts improve the spectral information content if field-induced broadening effects are not dominant. The overall quadratic dependence on the magnetic field of the thermal nuclear magnetization and the frequency of operation might largely compensate for the enhancement reduction generally observed at higher fields and, hence, allow to achieve a larger SNR. As a consequence, the previous two advantages make the exploration of DNP methods in high fields interesting and potentially impactful.

In this work, we designed and characterized single-chip integrated DNP microsystems operating at 200 GHz (ESR)/300 MHz (NMR). This effort is driven by the strong motivation to operate at higher static magnetic fields mentioned above. The realized single-chip microsystems allow to perform continuous wave ESR experiments by field or frequency sweeps and DNP-enhanced NMR experiments. Conventional high-field DNP methodologies require the use of relatively expensive and bulky microwave/quasi-optical sources and waveguides[43,45–47]. The integration of the microwave source and ESR detector together with the NMR sensor allows the study DNP-enhanced NMR without the need for external microwave components. The microwave generation and the microwave detection are performed in situ by the single chip microsystems integrated on an area of less than 1 mm². As a result, these single-chip integrated ESR, NMR, and DNP-NMR detectors are suitable for the miniaturization of the probe, for the reduction of the losses and complexity of the connections, and for the realization of dense arrays of detectors for parallel spectroscopy of several samples in the same magnet. By enabling simultaneous measurements, arrays of such sensors reduce the time and the cost of sample characterization. Additionally, the single-chip approach might allow for a better SNR for volume-limited samples in the nanoliter and sub nanoliter range tightly matched to the sensitive volume of the detector with respect to conventional bulky inductive probes optimized for microliter and larger sample volumes. By suppressing the need for external microwave sources and microwave connections, the single chip approach proposed here reduces drastically the cost and the complexity of the DNP instrumentation and, hence, should allow for more widespread use and study of DNP methodologies, particularly for nanoliter and subnanoliter samples.

## Results

In the following, we report on the design and characterization of two single-chip DNP microsystems consisting of co-integrated ESR and NMR detectors operating at about 200 GHz (ESR) and 300 MHz (NMR). The two chips, realized using a 130 nm SiGe technology (IHP SG13G2Cu), are shown in Fig. 1. The two NMR detectors differ in the diameter of the integrated microcoil (200 $\mu$m and 250 $\mu$m) whereas the two ESR detectors differ in the number of microwave oscillators integrated inside the NMR microcoil (one and four). The DNP microsystem with only one ESR oscillator is named single oscillator DNP microsystem, whereas the DNP microsystem with four ESR oscillators is named oscillator array DNP microsystem.

### Single oscillator DNP microsystem

The ESR detector consists of a 200 GHz voltage-controlled oscillator (VCO) based on differential Colpitts topology. Figure 2 shows the

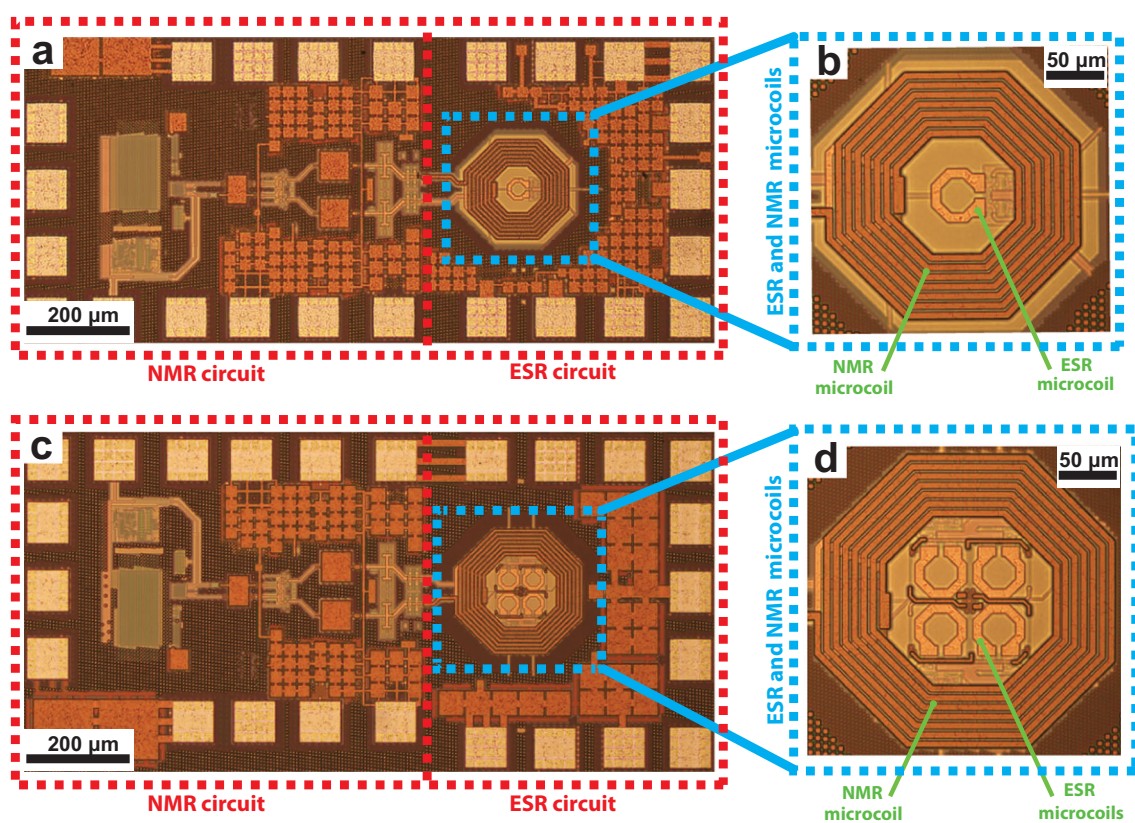

**Fig. 1 | Photographs of the single chip integrated DNP microsystems.** Both microsystems are realized using a SiGe technology and occupy a surface of about 1 mm². **a** Photograph of the single chip single oscillator DNP microsystem. The blue lines indicate the concentric NMR/ESR microcoils. The red lines indicate the NMR/ESR circuits. **b** Photograph of the NMR and ESR microcoils. The ESR microcoil, $L_{ESR}$, has a single turn and an external diameter of 45 $\mu$m. The NMR microcoil, $L_{NMR}$, has 21 turns (7 turns in each of the three top metal layers) and an external diameter of 200 $\mu$m. The total power consumption for the single oscillator DNP microsystem is in the range of 35 to 65 mW depending on biasing conditions. **c** Photograph of the single chip oscillator array DNP microsystem. **d** Photograph of the NMR and ESR microcoils. Each ESR microcoil has a single turn and an external diameter of 45 $\mu$m. The NMR microcoil has 21 turns (7 turns in each of the three layers) and an external diameter of 250 $\mu$m. The total power consumption for the oscillator array DNP microsystem is in the range of 50 to 175 mW depending on biasing conditions.

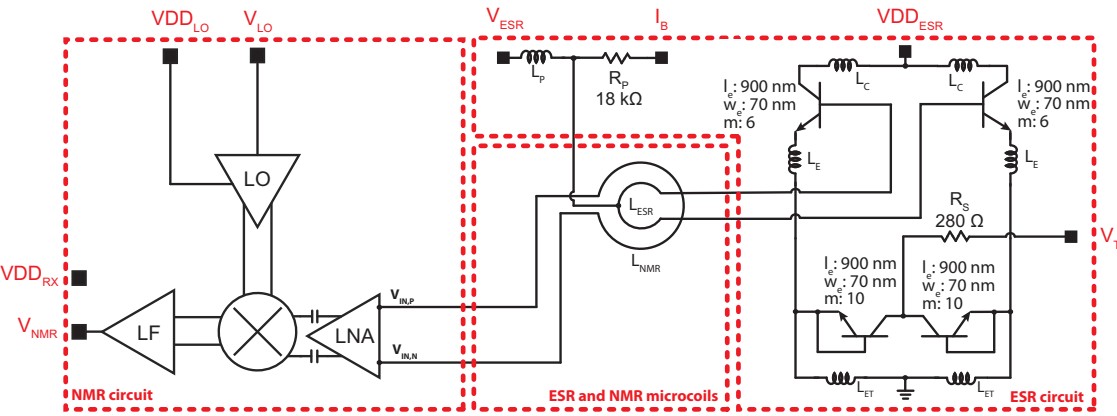

**Fig. 2 | Schematics of the single chip integrated DNP microsystem with a single oscillator.** The dashed red lines indicate the NMR/ESR circuits and the concentric NMR/ESR microcoils. The black squares next to the port names represent the bonding pads. On the left, the block diagram of the NMR circuit is given. $VDD_{RX}$ is the DC supply voltage of all blocks in the NMR receiver. It is connected to several nets and the connections are not shown in the schematic for simplicity. $VDD_{LO}$ is the supply voltage of the LO amplifier. $V_{LO}$ is the down converting local oscillator signal. $V_{NMR}$ is the output of the NMR detector. On the right, detailed schematics of the ESR oscillator are given. $l_e$ is the emitter length, $w_e$ is the emitter width, and $m$ is the multiplicity parameter of the heterojunction bipolar transistors. The inductor values are $L_C \cong 10$ pH, $L_E \cong 7$ pH, $L_{ET} \cong 50$ pH, $L_{ESR} \cong 60$ pH, and $L_P \cong 200$ pH. $VDD_{ESR}$ is the DC supply voltage of the oscillator. $I_B$ is the DC biasing current of the oscillator. $V_T$ is the tuning voltage of the oscillator. $V_{ESR}$ is the output of the ESR detector.

schematic design of the ESR circuit. The design of the oscillator we realized is based on the work by Voinigescu et al.[48]. The oscillator has been designed with the help of electromagnetic simulations (Advanced Design System, Keysight) combined with circuit simulations (Cadence). In particular, the simulations are performed by importing the technology model files provided by the foundry and the S-parameter files obtained from ADS simulations in Cadence. The oscillator operates with a supply voltage $VDD_{ESR}$ in the range from 1 to 1.5 V (with a $IDD_{ESR}$ current from 5 to 24 mA) and a supply current $I_B$ in the range from 6 to 70 $\mu$A (with a $V_B$ voltage from 1.15 to 2 V). The oscillator frequency can be varied up to 3 GHz by the tuning voltage $V_T$. The frequency tunability of the oscillator allows to perform continuous wave ESR and DNP-NMR experiments with frequency sweeps in addition to magnetic field sweeps[36]. The microwave magnetic field $B_{1e}$ generated by the oscillator at the center of the ESR microcoil is about 10 G (see Methods).

The ESR phenomenon produces a variation of the oscillation frequency and of the oscillation amplitude of the oscillator[35]. In principle, the output stage of the single-chip integrated oscillator could be electrically interfaced with an appropriate waveguide and the measurement of the frequency and amplitude variations caused by the ESR phenomenon could be performed outside of the magnet. However, due to the complexity and losses of such connections at high frequencies, the advantage of the single-chip approach is the possibility of using a frequency down-conversion circuit[31,35,49] or an amplitude detection circuit[50-52] integrated on the same chip of the oscillator. In this work, we used amplitude detection because of the simplicity of the required integrated circuitry. The DC component of the voltage at the $V_{ESR}$ node in Fig. 2 is proportional to the oscillation amplitude. This property had been shown in cross-coupled CMOS LC VCOs[50] and used for ESR detection[51,52]. The base of the Colpitts pair can be DC biased by a voltage or a current. The current bias $I_B$ of this node through the bias resistor $R_P$ allows for voltage variation at the center of the $L_{ESR}$ inductor and the $V_{ESR}$ node, and hence for the amplitude detection of the ESR signal. The amplitude-encoded ESR signal on the $V_{ESR}$ node is transferred out of the chip through an RF decoupling inductor $L_P$. Since the microwave magnetic field $B_{1e}$ is generated on chip by the oscillator and the ESR signal is encoded as a low-frequency signal at the field modulation frequency, no microwave connections are necessary (all signals from/to the single chip ESR detector are DC or low frequency).

The NMR detector is a receiver-only chain with 75 dB overall gain, consisting of a microcoil, a broadband low noise amplifier (LNA) operating up to 1 GHz, a mixer, and a low frequency (LF) amplifier with a 4 MHz bandwidth (see Fig. 2 and Supplementary Fig. 5). The NMR detection microcoil $L_{NMR}$ is realized with the three top metal layers available in the technology. It has an external diameter of 200 $\mu$m, a wire width of 5 $\mu$m, a wire thickness of 3 $\mu$m, a spacing between the wires of 2.8 $\mu$m, and a total number of turns of 21 with 7 turns in each layer. The simulated values for the inductance and the series resistance of the resulting NMR microcoil at the operating frequency of 300 MHz are 79 nH and 24 $\Omega$, respectively. Electromagnetic simulations show that an NMR microcoil diameter too tightly matched to the ESR microcoil diameter might not allow the ESR oscillator to work properly. For this reason, the NMR microcoil has an internal diameter of about 100 $\mu$m, a value which does not tightly match the 43 $\mu$m diameter of the ESR microcoil. The detailed circuit description for the NMR detector together with simulation results is presented in Supplementary Note 4. The measured deadtime is about 1 $\mu$s. The total input referred noise of the receiver is about 1 nV Hz$^{-1/2}$, as obtained by post layout and corner simulations. The measured spin sensitivity of the NMR receiver is about $3 \times 10^{13}$ $^1$H spins Hz$^{-1/2}$ at 300 MHz and room temperature (see Methods).

## Oscillator array DNP microsystem

The ESR microcoil of the single oscillator DNP microsystem excites efficiently a volume of about $50 \times 50 \times 25$ $\mu$m$^3$ (i.e., about 60 pL). For samples having volumes significantly larger than 100 pL, the efficient excitation of a larger volume would increase the NMR signal-to-noise ratio and the concentration sensitivity. Integrated arrays of NMR detectors[17,28] as well as ESR detectors[38,39,51] have been previously reported. In this work, we demonstrate an array of four frequency-locked oscillators for ESR excitation in a single-chip DNP microsystem. Having four oscillators coupled together increases the effective sample volume to about $100 \times 100 \times 50$ $\mu$m$^3$, i.e., about 0.5 nL, a factor of 8 larger with respect to the single oscillator microsystem.

Figure 1 shows a photo of the oscillator array DNP microsystem (c) and a photo of the ESR and NMR microcoils (d). The array system consists of four 200 GHz oscillators, capacitively coupled using the smallest capacitors (34 fF) available in the technology (see Supplementary Note 5 and Supplementary Fig. 7). The NMR receiver electronics are the same as the one of the single oscillator DNP

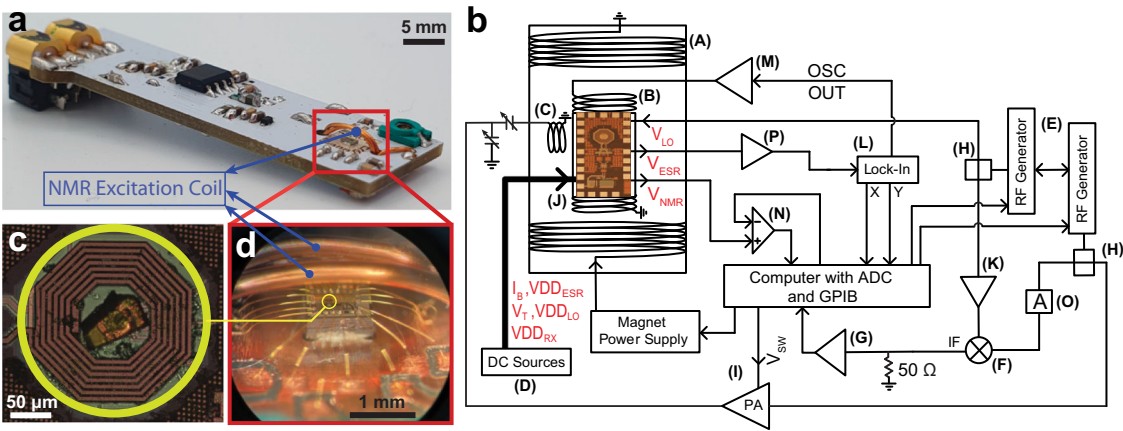

**Fig. 3 | Set-up for the characterization of the single chip integrated DNP microsystem operating at 200 GHz(ESR)/300 MHz(NMR). a** Printed circuit board (PCB) used for characterizing the single chip DNP microsystems. The red rectangle indicates the chip and the NMR excitation coil. **b** Block diagram of the complete setup for the characterization of the single chip DNP microsystems: (A) Super-conducting magnet with variable temperature insert (1.4 to 325 K, 0 to 9.4 T, Cryogenic Ltd); (B) Resistive modulation coil (2.8 mT/A); (C) NMR excitation coil with tuning and matching variable capacitors; (D) DC power supplies (Keithley 2400); (E) RF generators (Stanford Research Systems SG384); (F) Frequency mixer (Mini-Circuits ZAD-1); (G) Amplifier (Stanford Research Systems SR560); (H) Power splitters (Mini-Circuits ZSC-2-1+); (I) RF Power amplifier (RFPA). $V_{SW}$ is the control

signal connected to the power amplifier switch for generating pulsed excitations; (J) Photograph of the single chip DNP microsystem; (K) Amplifier (Mini-circuits ZX60-V83-S+); (L) Lock-in amplifier (EG&G 7265); (M) Power amplifier (Rohrer PA508); (N) Low noise differential amplifier (NF Corporation 5305); (O) 26 dB attenuator; (P) Non-inverting amplifier designed with an operational amplifier (THS4304D, Texas Instruments). The amplifier (P) is shown outside the magnet for a clear drawing. It is located on the PCB and placed inside the magnet together with the single chip microsystems. **c** Photograph of a 2%BDPA:PS sample placed on top of the ESR and NMR microcoils. **d** Photograph of the chip which shows the bonding wires and part of the NMR excitation coil.

microsystem, except for the NMR microcoil which is slightly larger. It has an external diameter of 250 μm (50 μm larger compared to the NMR microcoil of the single oscillator microsystem), and the same wire width, thickness, spacing and number of turns. The computed values for the inductance and the series resistance of the resulting NMR microcoil at the operating frequency of 300 MHz are 119 nH and 35 Ω, respectively. In this microsystem, the NMR microcoil has an internal diameter which is tightly matched to the chip area occupied by the ESR microcoil. In this design, contrary to the safer approach adopted for the single oscillator DNP microsystem described above, we took the risk of having a stronger perturbation of the ESR oscillators by the NMR microcoil. Experimentally we observe that also the array of coupled ESR oscillators works properly, with a slightly lower frequency and a similar power consumption per oscillator. The frequency of the oscillator array is tunable by approximately 1 GHz by changing the tuning voltage $V_T$ from 0 to 2 V. The oscillator array has an operating frequency which is approximately 5 GHz lower than the single oscillator. The measured NMR spin sensitivity of the array microsystem is about $4 \times 10^{13}$ spins Hz$^{-1/2}$, i.e., slightly worse than the one of the single oscillator microsystem due to the larger NMR microcoil (see Methods).

**Experimental setup**

The measurement setup is shown in Fig. 3. The single-chip DNP microsystems are glued on a printed circuit board (PCB) with conductive epoxy (Epo-Tek, H20E-FC), and the connections from the chip to the PCB are made by gold bonding wires. Figure 3a shows the complete PCB and Fig. 3d shows the chip together with the bonding wires. The PCB is inserted into a superconducting magnet with variable temperature insert (0 to 9.4 T, 1.4 to 325 K, Cryogenic Ltd.) having a bore diameter of 28 mm. Two coaxial cables and twelve twisted pair wires are used to connect the electronics on the PCB to the external electronics. One coaxial cable is used for the NMR excitation signal. The other coaxial cable carries the local oscillator signal $V_{LO}$ that is used in the on-chip frequency down conversion of the NMR signal. All the other connections, including those for the amplified and down-converted NMR signal and the ESR signal, are made using the twisted wires.

The ESR spectra are recorded by measuring the oscillation amplitude variation as a function of the oscillator frequency or the static magnetic field. Conventional magnetic field modulation and lock-in demodulation are used to improve the signal-to-noise ratio. The ESR signal at the output of the chip (i.e. at the node $V_{ESR}$ in Fig. 2) is amplified by a non-inverting amplifier designed with an operational amplifier (THS4304D, Texas Instruments) located on the PCB (Fig. 3a). This operational amplifier is chosen because it is known to operate also at cryogenic temperatures[34]. A resistive feedback with capacitors is used to set the gain to 100 V/V and the bandwidth from 1 to 150 kHz. The passband filter determines the field modulation frequency range in the continuous wave ESR experiments.

All NMR and DNP-enhanced NMR experiments reported in this work are conventional free induction decay (FID) measurements performed after a single π/2 pulse. The microwave excitation from the integrated ESR oscillator is permanently on during all measurements. The pulse sequence diagram is given in Supplementary Note 3. The NMR excitation coil is tuned with a series-connected varactor (Sprague-Goodman SGC3S100NM) and matched with a parallel connected varactor (two Sprague-Goodman SGC3S300NM with a 10 pF capacitor in parallel) as shown in Fig. 3a, b. The excitation coil has two turns, a diameter of 6.5 mm and a wire diameter of 400 μm which results in 50 nH inductance and 1 Ω series resistance. The NMR output, $V_{NMR}$, is taken directly out of the chip and not further amplified on the PCB. Outside of the magnet, it is further amplified (amplifier (N) in Fig. 3b). Since the $V_{NMR}$ has a DC component of about 1 V, a differential amplification with a DC voltage source set to 1 V is required. Two frequency locked RF generators are used for NMR excitation and LO down conversion. Even though they are frequency locked at a fixed frequency difference (125 kHz in our experiments), phase lock is not guaranteed. Since phase coherence is necessary for NMR signal averaging, a reference signal is generated by mixing these two RF signals (excitation and $V_{LO}$, with the mixer (F) in Fig. 3b). The NMR signal and the reference signal are digitized by a 16 bits analog-to-digital converter operating with a sampling frequency of 700 kHz per channel. The phase correction of the NMR signal is based on the measurement of the phase of the reference signal and performed by the Labview$^{TM}$

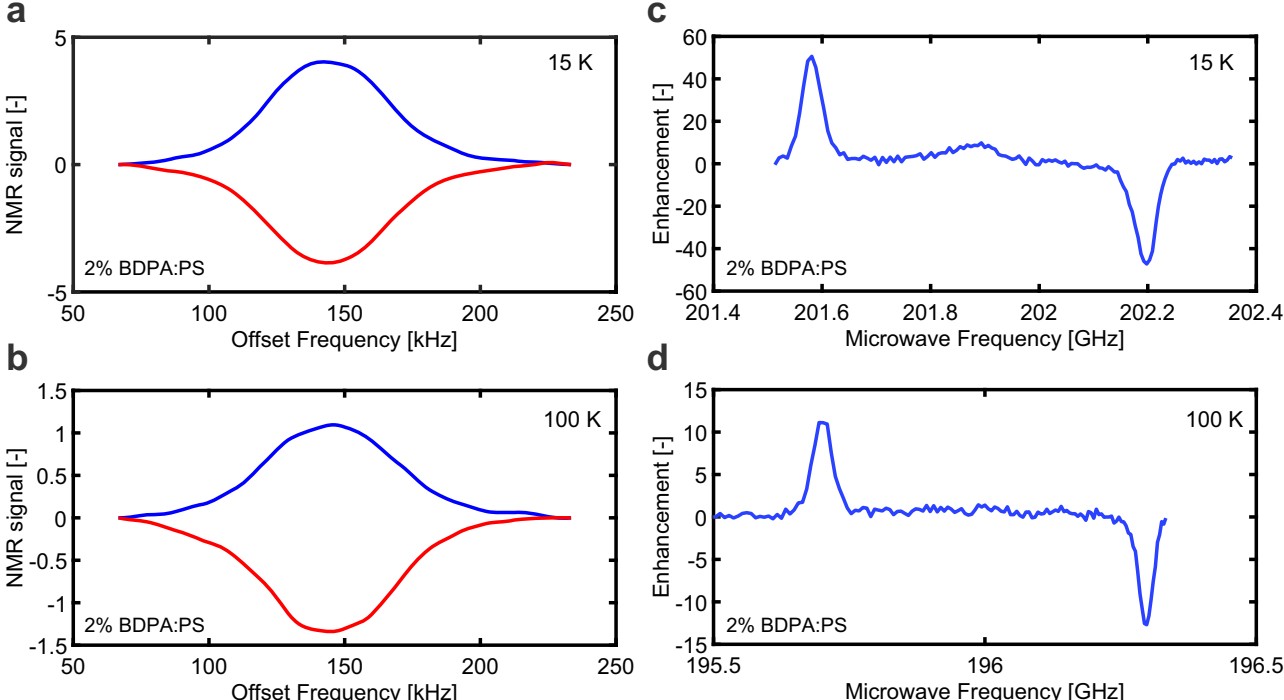

**Fig. 4 | $^1$H NMR spectra and DNP-enhancement curves measured with the single chip integrated single oscillator DNP microsystem. a** $^1$H NMR spectra and **c** DNP-enhancement curve measured with a sample of 2% BDPA in PS having a volume of $100 \times 200 \times 100$ $\mu m^3$ at 15 K. The oscillator is supplied with $I_B = 30$ $\mu A$, VDD$_{ESR}$ = 1.1 V and IDD$_{ESR}$ = 19 mA. The NMR measurements are performed in the following conditions: $f_{rf} \cong 307$ MHz, pulse length $\tau_{rf}$ = 3 $\mu s$, pulse repetition time $T_r$ = 4 s, time-domain match filter time constant $T_m$ = 30 $\mu s$, acquisition time $T_{daq}$ = 15 ms, number of averaging $N_{avg}$ = 80. In **a** the blue curve is the DNP-enhanced NMR spectra when the microwave frequency is $\omega_{OS} - \omega_{OI} \cong 201.6$ GHz (i.e., to the left of the ESR signal)

and the red curve is the DNP-enhanced NMR spectra when the microwave frequency is $\omega_{OS} + \omega_{OI} \cong 202.2$ GHz (i.e., to the right of the ESR signal). **b** $^1$H NMR spectra and **d** DNP-enhancement curve measured with a sample of 2% BDPA in PS having a volume of $100 \times 50 \times 25$ $\mu m^3$ at 100 K. The oscillator is supplied with $I_B = 45$ $\mu A$, VDD$_{ESR}$ = 1.05 V and IDD$_{ESR}$ = 23 mA. The NMR measurements are performed in the following conditions: $f_{rf} \cong 297$ MHz, $\tau_{rf}$ = 11 $\mu s$, $T_r$ = 10 s, $T_m$ = 30 $\mu s$, $T_{daq}$ = 15 ms, $N_{avg}$ = 40. In **b** the blue curve is the DNP-enhanced NMR spectra when the microwave frequency is $\omega_{OS} - \omega_{OI} \cong 195.7$ GHz and the red curve is the DNP-enhanced NMR spectra when the microwave frequency is $\omega_{OS} + \omega_{OI} \cong 196.3$ GHz.

program which controls the entire experiment. The frequency of the RF generator connected to the excitation coil is set close to the NMR resonance for optimal on-resonance excitation. The frequency generator used for the LO signal is set at a frequency which is 125 kHz above or below the excitation frequency. With this approach, the downconverted NMR signal is at about 125 kHz. This approach reduces significantly the spectral distortions also for NMR lines as large as 40 kHz and avoids low-frequency noise and interferences.

## DNP experiments with the single oscillator microsystem

To exemplify the possible applications of the 200 GHz single chip DNP microsystem demonstrated in this work, in the following we report on DNP experiments on solid samples performed in the temperature range from 15 to 300 K. Figure 4 shows $^1$H NMR spectra (left column) and DNP-enhancement curves (right column) measured with a sample of 2% $\alpha,\gamma$-bisdiphenylene-$\beta$-phenylallyl (BDPA) in polystyrene (PS) having a volume of $100 \times 200 \times 100$ $\mu m^3$ and $100 \times 50 \times 25$ $\mu m^3$ at 15 K and 100 K. Both DNP curves are measured by sweeping the frequency of the oscillator by V$_T$. At each V$_T$ value, the $^1$H NMR spectra are measured with 80 and 200 averaging at 15 and 100 K, respectively. At 15 K, both the solid effect and the Overhauser effect are observed (see Fig. 4c), as previously reported[53]. In the solid effect conditions (i.e., for $\omega_{OS} \pm \omega_{OI}$ where $\omega_{OS} = \gamma_e B_0$ is the ESR frequency and $\omega_{OI} = \gamma_I H B_0$ is the NMR frequency), the NMR signal is enhanced by approximately ±50 and ±10 times at 15 and 100 K, respectively. These results are similar to those of previous works. Enhancements as large as 12 at 9.4 T and 100 K[53] and 10 at 5 T and 300 K[54] have been reported for the same sample. Enhancements of 8 at 300 K and 15 at 100 K at 18.8 T have been reported for a highly deuterated PS sample doped with BDPA[55].

## DNP experiments with the oscillator array microsystem

Figure 5 shows $^1$H NMR spectra (left column) and DNP-enhancement curves (right column) measured with a sample of 2% BDPA in PS having a volume of $100 \times 175 \times 50$ $\mu m^3$ at 200 K and samples of 2% BDPA in SEBS having volumes of $90 \times 90 \times 30$ $\mu m^3$ and $200 \times 200 \times 100$ $\mu m^3$ at 300 K. All four DNP curves are measured by sweeping the frequency of the oscillator. The microwave frequency is set by the tuning voltage $V_T$ of the oscillator. At each microwave frequency, the $^1$H NMR spectrum is measured. The non-enhanced NMR signals for 2% BDPA:SEBS samples are shown in green in Fig. 5b–d. To the best of our knowledge, DNP experiments on BDPA:SEBS samples are performed for the first time in this work. An Overhauser DNP is observed at room temperature in Fig. 5f–h. Overhauser and solid effect enhancements are below two at 300 K, whereas the solid effect enhancements increase up to 20 at 200 K. Overhauser enhancements in BDPA:PS have been observed in previous works[53].

The DNP-enhancement curves shown in Fig. 5f–h are measured with the same sample, while the sample size of Fig. 5h is 16 times larger. The oscillator is biased at the same values, whereas the repetition time $T_r$ is longer for the curve in Fig. 5f. The Overhauser enhancements are similar for both DNP curves in Fig. 5f and g (about 1.2) whereas the solid effect enhancement is larger for Fig. 5f compared to the Fig. 5g (about 1.7 and 1.45, respectively). This is most probably due to the longer build up time of the solid effect compared to the Overhauser effect. The Overhauser enhancements are also similar for both DNP curves in Fig. 5g, h whereas the solid effect enhancement is larger for Fig. 5g compared to Fig. 5h (about 1.45 and 1.05, respectively). This behavior is probably due to the Overhauser effect being easier to saturate with respect to the solid effect[53]. Since for the larger sample the average

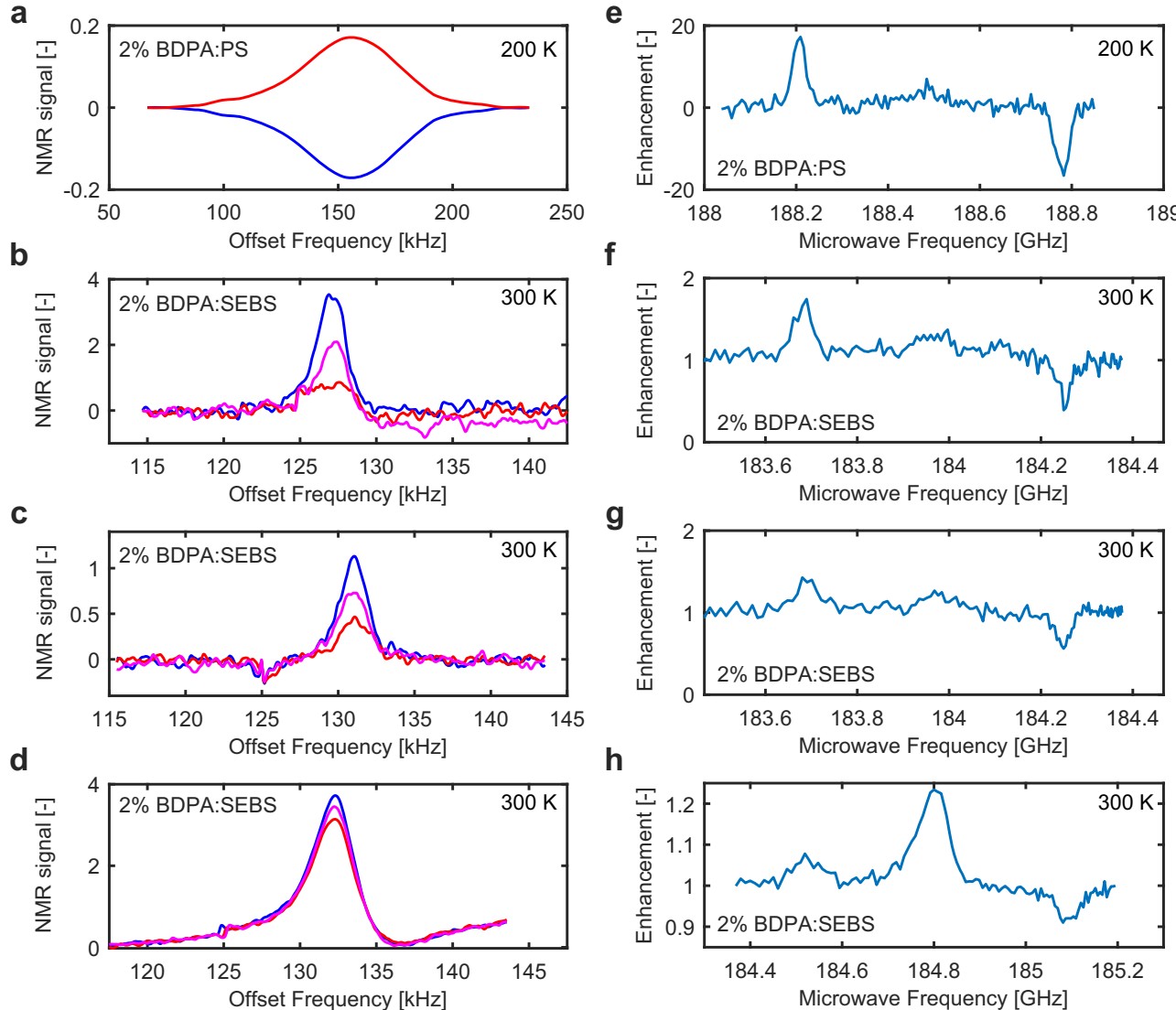

**Fig. 5 | ¹H NMR spectra and DNP-enhancement curves measured with the single chip integrated oscillator array DNP microsystem. a** ¹H NMR spectra and **e** DNP-enhancement curve measured with a sample of 2% BDPA in PS having a volume of $100 \times 175 \times 50~\mu m^3$ at 200 K. The oscillators are supplied with $I_B = 180~\mu A$, $VDD_{ESR} = 1.5$ V and $IDD_{ESR} = 80$ mA. The NMR measurements are performed in the following conditions: $f_{rf} \cong 287$ MHz, $\tau_{rf} = 8~\mu s$, $T_r = 500$ ms, $T_m = 30~\mu s$, $T_{daq} = 15$ ms, $N_{avg} = 720$. **b** ¹H NMR spectra and **f** DNP-enhancement curve measured with a sample of 2% BDPA in SEBS having a volume of $90 \times 90 \times 30~\mu m^3$ at 300 K. The oscillators are supplied with $I_B = 140~\mu A$, $VDD_{ESR} = 1.1$ V and $IDD_{ESR} = 60$ mA. The NMR measurements are performed in the following conditions: $f_{rf} \cong 280$ MHz, $\tau_{rf} = 18~\mu s$, $T_r = 1$ s, $T_m = 1$ ms, $T_{daq} = 15$ ms, $N_{avg} = 480$. **c** ¹H NMR spectra and **g** DNP-enhancement curve measured with a sample of 2% BDPA in SEBS having a volume of $90 \times 90 \times 30~\mu m^3$ at 300 K. The oscillators are supplied with $I_B = 140~\mu A$, $VDD_{ESR} = 1.1$ V and $IDD_{ESR} = 60$ mA. The NMR measurements are performed in the following conditions: $f_{rf} \cong 280$ MHz, $\tau_{rf} = 10~\mu s$, $T_r = 100$ ms, $T_m = 1$ ms, $T_{daq} = 15$ ms, $N_{avg} = 3600$. **d** ¹H NMR spectra and **h** DNP-enhancement curve measured with a sample of 2% BDPA in SEBS having a volume of $200 \times 200 \times 100~\mu m^3$ at 300 K. The oscillators are supplied with $I_B = 140~\mu A$, $VDD_{ESR} = 1.1$ V and $IDD_{ESR} = 60$ mA. Each NMR measurement is performed in the following conditions: $f_{rf} \cong 281$ MHz, $\tau_{rf} = 8~\mu s$, $T_r = 100$ ms, $T_m = 1$ ms, $T_{daq} = 15$ ms, $N_{avg} = 3600$. In **a–d** the blue curve is the DNP-enhanced NMR spectra when microwaves are applied on the left side of the ESR peak ($\omega_{OS} - \omega_{OI}$), the red curve is the DNP-enhanced NMR spectra when microwaves are applied on the right side of the ESR peak ($\omega_{OS} + \omega_{OI}$). In **b–d** the green curve is the NMR spectra when microwaves are far from the ESR peak. The tiny signal at 125 kHz is a parasitic signal corresponding to the frequency difference between the LO signal and the RF excitation signal.

microwave magnetic field is weaker, we would expect that the ratio between the Overhauser and the solid effect is increased in the larger sample, which is consistent with our experiments.

## DNP-enhancement estimation

The NMR excitation coil is used to excite the nanoliter samples placed on top of the integrated NMR microcoil. However, this coil excites efficiently also a volume of about 50 $\mu$L in the FR4 epoxy PCB, a material with a high density of ¹H nuclei. Despite the presence of a Cu layer having a thickness of 37 $\mu$m between the excited volume in the PCB and the integrated detection coil, the NMR signal induced in the

detection coil by the ¹H nuclei in PCB substrate can be larger than the non-enhanced NMR signal induced in the detection coil by ¹H nuclei contained in a sample having a volume significantly smaller than the sensitive volume of the detection coil. In order to compute the DNP-enhancement, we quantified the background NMR signal by performing measurements without a sample on top of the detection coil. For 2% BDPA:PS, the background NMR signal is much larger than the non-enhanced NMR signal from the sample. This large ratio makes an accurate experimental evaluation of the non-enhanced NMR signal difficult because the background NMR experiment and the experiment in the presence of the sample cannot, in practice, be performed in the

exact same conditions and are affected by noise. In our experimental conditions, we estimated that the background signal cannot be determined with an error of less than 10%. In experiments where the non-enhanced NMR signal is below 10% of the background signal, we assumed that the non-enhanced NMR signal is 10% of the background signal. To cross-check this assumption, we calculated the expected NMR signal from the sample using a C-program based on the solution of the Bloch differential equations, the reciprocity principle, and the geometry of the sample and of the excitation and detection coils[56]. The computed NMR signal is smaller than the 10% of the background signal, which is the NMR signal amplitude used in the enhancement calculations. Therefore, this criterion leads to an underestimation of the DNP enhancement.

It is also important to notice that DNP-enhancement values reported in this work are not normalized considering the ratio between the total sample volume and the volume of the sample efficiently excited by the ESR microcoil. In principle, electromagnetic simulations combined with the obtained experimental results could be used to obtain a DNP magnetization enhancement map inside the sample. However, this would require accurate knowledge of the electromagnetic properties at 200 GHz of all materials involved and a hypothesis on the DNP enhancement vs $B_{1e}$ relation. For this reason, we prefer to report the globally observed DNP enhancements, knowing that, for samples having a volume significantly larger than the volume efficiently excited by the ESR microcoil, these enhancement values are an underestimation of the DNP-induced NMR magnetization enhancement in close proximity to the ESR microcoil.

## Discussion

In this work, we demonstrated the possibility of extending the single-chip approach to the realization of probes for DNP studies of nanoliter and subnanoliter samples at magnetic fields up to 7 T and temperatures down to 15 K. The single chip DNP microsystem consists of a 200 GHz oscillator (as an ESR detector) and a broadband receiver operating up to 1 GHz (as an NMR detector). The ESR excitation and the ESR amplitude detection are performed on-chip without the need for external microwave sources and microwave connections. Measurements on 2% BDPA:PS sample show solid effect enhancements as large as 50 and 10 on samples having an effective volume of about 2 nL and 125 pL at 15 K and 100 K, respectively. A single-chip DNP microsystem with an array of four 190 GHz oscillators (i.e. oscillator array DNP microsystem) is also reported to demonstrate the possibility of efficiently exciting a larger volume. Measurements performed with this DNP microsystem on a 2% BDPA:PS sample show enhancements as large as 20 on samples having an effective volume of about 1 nL at 200 K. Both microsystems are integrated on a single chip having an area of less than 1 mm².

In the following, we discuss possible improvements and extensions of the high-frequency single-chip DNP microsystem approach demonstrated in this work. The experiments performed on solid samples having short $T_2^*$ suffered from the presence of a large NMR background signal having a similar linewidth. The background signal is generated by the protons contained in the PCB supporting the single chip integrated microsystem. The single-chip integrated microsystem is made of materials that do not contain protons. To drastically reduce the background signal, hence allowing for a more accurate determination of the DNP enhancement, PCBs that do not contain ¹H nuclei, such as those based on alumina or polytetrafluoroethylene (PTFE), could be used. Another possible solution to the background signal problem is the co-integration on a single chip of the NMR excitation coil, the NMR transmitter, and the NMR receiver, as previously reported in a single chip 10 GHz DNP microsystem[40]. The localized NMR excitation produced by integrated microcoil drastically reduces the contribution to the measured signal from the protons contained in the PCB.

The use of state-of-the-art submicrometer integrated circuit technologies should allow the extension of the single-chip DNP microsystem approach proposed here up to the 1 THz (ESR)/1 GHz (NMR) region[57–60], corresponding the strongest static magnetic fields currently available.

Pseudo-pulsed DNP experiments could be performed using the frequency tunability of the single chip integrated oscillators. An experimental scheme to perform pulsed ESR measurements with a voltage controlled oscillator (VCO) is reported in ref. 61.

The microwave field produced by the microcoils of the integrated oscillators can be changed by less than a factor of two by a variation of the oscillator supply voltage $VDD_{ESR}$ and oscillator current $I_B$. The minimum microwave field is determined by the start-up condition of the oscillator, whereas the maximum microwave field is limited by the maximum supply voltage that we can apply to the oscillator (about 1.5 V). In the future, a larger variation of the effective microwave field might be achievable by a different oscillator design or by pulse modulation with a variable duty cycle of the microwave frequency from resonance to out-of-resonance conditions using the oscillator tuning voltage $V_T$.

An interesting opportunity offered by the single-chip integration approach is the possibility of creating dense arrays of such sensors for parallel DNP-enhanced NMR spectroscopy of a large number of nanoliter and subnanoliter different samples (or a bigger volume of the same sample for enhanced concentration sensitivity). Previously, arrays of integrated ESR detectors[38,39,51] and NMR detectors[17,28] have been published. In this work, a single-chip DNP array microsystem consisting of four frequency-locked oscillators operating at about 200 GHz is reported. Recently, an array of 32 frequency-locked oscillators operating at 263 GHz was reported[38].

DNP experiments on solid samples are often performed with magic angle spinning (MAS) to improve the spectral resolution. A possible approach for incorporating MAS capabilities into the single-chip DNP microsystem involves rotating the chip concurrently with the sample. This would require the design of an efficient inductive coupling scheme to produce the necessary DC bias voltages and to transfer the RF input and output signals. A similar approach has been previously demonstrated, although in the simpler case of a spinning microcoil inductively coupled with a static coil connected to the probe electronics[62]. A second possible solution would be the optical MAS of the samples in the sensitive volume of the single-chip DNP microsystem. Optical MAS might allow to exceed the spinning frequency limit of the current MAS techniques by an order of magnitude. Preliminary experiments in this direction are reported in ref. 63 where the experiments on the optical levitation and rotation of subnanoliter samples are reported but the possibility to perform NMR/ESR/DNP experiments on such samples in non-static conditions is still to be demonstrated.

NMR experiments on subnanoliter biological samples using single chip NMR detectors have been recently reported[24,26,27,64]. An interesting extension would be the use of single-chip DNP microsystems for such experiments with the addition of suitable biocompatible polarizing agents. For these experiments, the samples have to be confined into microfluidic reservoirs, ideally at a distance of less than 20 $\mu$m from the chip surface for efficient ESR excitation with the integrated ESR microcoils. Such microfluidic devices can be fabricated by high-resolution 3D printing[26] for NMR experiments on subnanoliter samples. As for all previously reported high field in-situ DNP-enhanced NMR experiments on liquid samples, the challenging aspects are related to the relatively poor DNP enhancements demonstrated so far combined with possible significant microwave heating of samples in aqueous media.

## Methods

### ESR microcoil design

Electromagnetic (Advanced Design System, Keysight) and electronic circuit simulations (Cadence) indicate that, due to parasitic

capacitances, operation at 200 GHz requires an ESR microcoil having a diameter not much larger than 40 $\mu$m. The ESR microcoil is designed using the second metal layer from the top. The first layer is thicker than the second layer, but its sheet resistance is similar (the top layer is made of aluminum whereas all other layers are made of copper). Moreover, the connection from the second layer to the first layer needs to be done through a resistive thin layer and resistive vias. The third layer has the same sheet resistance as the second layer. However, the use of the third layer would increase the effective distance between the microcoil and the sample by about 9 $\mu$m, hence slightly decreasing the microcoil sensitivity for a microcoil having a diameter of only 40 $\mu$m. The vias from the third layer to the second layer have a sufficiently low resistance to have a negligible impact on the oscillator performance. These considerations justify the choice of the second metal layer for the ESR microcoil. The ESR microcoil $L_{ESR}$ consists of a single-turn planar coil having an external diameter of about 43 $\mu$m, a wire width of 7 $\mu$m, and a wire thickness of 3 $\mu$m. The simulated values for its inductance and series resistance at the operating frequency of 200 GHz, including the connections to the transistor pins, are 60 pH and 8 $\Omega$, respectively.

## ESR oscillator operation frequency range

The oscillator frequency can be varied by the tuning voltage $V_T$. The standard varactors available in the chosen technology have too large values for the target frequency of 200 GHz. For this reason, two transistors are used as variable capacitors in base-emitter short connection (see Fig. 2), as suggested in ref. [48]. In order to determine the oscillator frequency and the frequency tunability we performed experiments with a sample of $\alpha,\gamma$-bisdiphenylene-$\beta$-phenylallyl (BDPA, 152560, Sigma-Aldrich) having a volume of a few $\mu$m$^3$. The operating frequency of the integrated oscillator is determined assuming an effective gyromagnetic ratio for BDPA of 28.02 GHz/T[65]. The magnetic field of the superconducting magnet with variable temperature insert (1.4 to 325 K, 0 to 9.4 T, Cryogenic ltd.) is measured using $^1$H NMR measurements (the absolute accuracy of such measurements is in the order of 10 ppm). The frequency of the single oscillator can be changed up to 3 GHz by changing the tuning voltage $V_T$ between 0 and 2 V. An additional 5 GHz tuning is achieved by adjusting the supply current $I_B$ and the supply voltage VDD$_{ESR}$. Hence, the overall frequency tuning range of the oscillator is about 8 GHz. The oscillator frequency also depends on the operating temperature. The frequency increases approximately linearly as the temperature decreases, with a central frequency of 190 GHz at room temperature and 200 GHz at 10 K. The frequency of the oscillator array is tunable by approximately 1 GHz by changing the tuning voltage $V_T$ from 0 to 2 V. The frequency tunability by changing $I_B$ and VDD$_{ESR}$ is of about 5 GHz, as for the single oscillator. Hence, the overall frequency tuning range of the oscillator array is about 6 GHz.

## Microwave magnetic field ($B_{1e}$) estimation

To estimate the microwave magnetic field produced by the integrated oscillator we performed continuous wave ESR measurements on a BDPA sample having a volume of a few $\mu$m$^3$. The small sample volume is required to avoid the strong coupling regime[66]. Assuming a Lorentzian line shape, the full width at half maximum of the resonance line (in Hz) measured in a continuous wave experiment is $(1/\pi T_2)(1 + \gamma_e^2 B_{1e}^2 T_1 T_2)^{1/2}$. Hence, from the measurement of the linewidth and assuming $T_1 \cong T_2 \cong 100$ ns for BDPA[67,68], it is possible to determine the microwave magnetic field $B_{1e}$ produced by the oscillator microcoil at the position of the sample. The measured maximum $B_{1e}$ value at the center of the ESR microcoil is of about 10 G, in good agreement with simulations showing a maximum $B_{1e}$ of about 5 G.

The microwave magnetic field produced by the array of four oscillators is estimated by a combination of ADS and Cadence simulations as for the one produced by the single oscillator. This simulation shows that the microwave current running in each of the four microcoils is almost identical to the one running in the single oscillator microcoil. Since the four oscillators are frequency and phase locked, the four microcoils behave as one larger microcoil (see Supplementary Note 5). The diameter of this larger microcoil is approximately two times larger than the one of the single microcoil. Hence, the microwave magnetic field produced by the array of four oscillators is approximately a factor two smaller compared to the one produced by the single oscillator.

## NMR spin sensitivity

In order to measure the $^1$H spin sensitivity of the NMR receiver, we used a solid sample of polyisoprene (Aldrich, 182141) having a volume of about 1 nL placed at the center of the NMR microcoil. The $^1$H spin density of polyisoprene is $6.4 \times 10^{28}$ spins m$^{-3}$, as computed from the density of 9100 kg m$^{-3}$ and the elemental composition (C$_5$H$_8$)$_n$ of polyisoprene. The measured spin sensitivity of the NMR receiver is about $3 \times 10^{13}$ $^1$H spins Hz$^{-1/2}$ at 300 MHz and room temperature. It is calculated as $N_{min} = V_s N_s \text{VSD}/S_O$ where $V_s$ is the sample volume (in m$^3$), $N_s$ is the spin density (in spins m$^{-3}$), VSD is the voltage noise spectral density (in V Hz$^{-1/2}$) and $S_O$ is the NMR signal amplitude at the beginning of the free induction decay (in V), both measured at the output of the chip. In a DNP-enhanced NMR experiment, the spin sensitivity is improved by the DNP-enhancement factor.

## Sample preparation

In order to characterize the performance of the realized single chip DNP microsystems, we prepared polystyrene (PS) and styrene-ethylene-butylene-styrene (SEBS) solid polymers doped with 2% $\alpha,\gamma$-bisdiphenylene-$\beta$-phenylallyl (BDPA/benzene, 152560, Sigma-Aldrich). The ESR linewidth of BDPA in these polymers is in the order of 10 G (28 MHz), hence it is suited for observing well separated solid effect (SE) lines in the DNP-enhancement profile at the $^1$H Larmor frequency of 300 MHz, corresponding to a field offset for the solid effect of about $\pm$110 G. BDPA:PS has been investigated in several previous DNP works[53,54,69,70] whereas BDPA:SEBS has not been investigated yet. At room temperature, the $^1$H NMR linewidth of PS is of about 50 kHz whereas the one of SEBS is of about 1 kHz, presumably due to motional narrowing as observed also in other elastomers above their glass temperature[17,22,56]. The narrower linewidth of the SEBS sample enables to obtain a larger signal-to-noise ratio and, hence, to perform faster measurements at room temperature.

The samples of 2% BDPA:PS and 2% BDPA:SEBS are prepared by dissolving PS (331651, MW 35000, Sigma-Aldrich) and BDPA in chloroform and SEBS (Kraton Corporation) and BDPA in toluene, respectively. The mixture is then drop-casted on a glass plate and dried overnight in a fumehood with a glass cover to reduce the speed of evaporation. If the evaporation is too fast, the top surface of the sample becomes very irregular. Once the sample is completely dried, the BDPA:PS samples are cut by a ceramic blade to obtain the desired dimensions. The BPDA:SEBS samples are difficult to cut due to their stretchability. In order to overcome this problem, SEBS samples are cut using an optical laser (Optec LightShot LSV3). Measurements show that the laser does not deteriorate the samples. Once cut, the samples are picked up by toothpicks sharpened by a cutter. The samples adhere to the toothpick by Van der Waals force. Once in contact with the chip surface, the samples adhere to it, also by Van der Waals forces. Although not strictly necessary, a small amount of paper glue (Pritt, Henkel AG) is sometimes used to strengthen the adhesion. As an example, in Fig. 3c, a 2% BDPA:PS sample on top of the sensitive area of the microchip (i.e., on top of the concentric ESR and NMR microcoils) is shown.

## Reporting summary

Further information on research design is available in the Nature Portfolio Reporting Summary linked to this article.

## Data availability

All data that support the key findings in this study are available within the main text and the Supplementary Information file. Source data are provided with this paper. Additional raw data are available from the corresponding author upon request. Source data are provided with this paper.

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

## Acknowledgements

Financial support from the Swiss National Science Foundation (SNSF) is gratefully acknowledged (grant 200020-175939). We thank Sami Jannin and Andrea Capozzi for valuable discussions, and Roberto Russo for his help in laser cutting of the samples.

## Author contributions

G.B. conceived the idea, supervised the research, contributed to experiments. N.S.S. and R.F. designed the chip, contributed to experiments. N.S.S. prepared the samples. N.S.S., R.F. and G.B. co-wrote the manuscript.

## Competing interests
The authors declare no competing interests.
