## [Peer Review File · Nature Communications]

200 GHz single chip microsystems for dynamic nuclear polarization enhanced NMR spectroscopyREVIEWER COMMENTS

Reviewer #1 (Remarks to the Author):

Review report

Title: 200 GHz single chip dynamic nuclear polarization microsystems
by Nergiz Sahin Solmaz et al.
submitted to Nature Comm.

This paper reports a new chip that has integrated both RF and microwave circuitry to demonstrate a micro-NMR system for performing DNP-enhanced NMR experiments. The authors have provided an excellent introduction to motivate the study, described the design of both the RF and microwave systems in some detail, and demonstrated the excellent performance of their systems. The enhancement appears to be stronger at lower temperatures, indicating the limited microwave power from the chip. I agree with the authors that such on-chip integration of NMR and ESR circuitry could help reduce the complexity of DNP-NMR and make it available for broader applications. Thus, I support the publication of this work after consideration of the following minor comments.

Title should reflect the full NMR system, instead just the DNP part. For example, "200 GHz single chip dynamic nuclear polarization micro-NMR system"

Please include simulations or measurements of the isolation between the RF coil and the microwave coil, perhaps in the supplementary information. Also LNA gain and noise spectrum plots.

On figure 2, please clarify the use of the V_{ESR} and I_B .

Explain how the LO amplifier (Fig 2b) produces different phases at LO_p and LO_n.

Figure 4, what are the red and blue lines? It would be great to include a pulse sequence diagram to clarify how the experiments were performed.

For the DNP experiment, the paragraph starting from line 195, please explain the choice of samples and what is the source of ESR.

It would be useful to demonstrate the microwave power dependence of the NMR enhancement as a way to estimate the degree of saturation of ESR in the current system.

Reviewer #2 (Remarks to the Author):

The manuscript describes the successful design and construction of two single chip microsystems capable to perform dynamically nuclear polarized magnetic resonance on solids doped with paramagnetic species.

Overall, the manuscript is of very high quality and in my eyes, it is a tour de force in microelectronics combined with advanced magnetic resonance. Achieving this kind of sensitivity a miniaturized integration deserves publication. Given the broad audience of Nature communications I would suggest the authors to revise this document and address some of the following points:

1) DNP is by no means a generic technique and even though a lot of efforts have been placed to push it the need for unpaired electrons in the sample reduces its applicability. Is there a benefit going microscopic for this drawback of DNP ?

2) The motivations of the authors to develop these high field microsystems is claimed to be: 1) better resolution, 2) higher thermal polarization. In my eyes both arguments are not valid: 1) DNP experiments in higher and higher magnetic fields have until now given more problems than solved (broadening due to disorder increases thus resolution is heavily impacted in poorly crystalized solids and amorphous glasses, heating becomes an issue, more expensive and specialized

hardware, lower enhancements, etc.). 2) Given that the polarization comes from the electrons increasing the field has much less effect (linear scaling) compared to reducing the temperature (nonlinear). Therefore, the main valid arguments I see is cost and the possibility to measure many miniature samples (maybe even in parallel with arrays?). The authors should rework a bit these points and make a more detailed motivation list with pros and cons in their final manuscript version.

3) The document is highly technical and for the general reader hard to follow. Maybe some technical details that are not necessary for following the flow of the work could be added in Supplementary material? Maybe in these SI they could also add more detailed drawings in case someone wished to reproduce this work (there were no financial interests disclosed).

4) Figure 2: I_e , w and m are not defined as far as I can see and have nm units.

5) Some parts of the text are repetitive (page 5 for example, top paragraph).

6) Background from protons seems to be a big issue, but no sophisticated background suppression techniques from NMR were developed/applied. The NMR user is going to wonder if these are possible with these MEMS or not and why. The authors should explain a bit more in detail.

7) The fact that the samples are static and not spinning at the magic angle limits tremendously the applicability of these devices towards conventional solid-state NMR studies. The authors mention inductive coupling under MAS as a solution without any references to working examples/experiments. Similarly the reference to the optical MAS is cryptic, hard to find and could be possibly updated to [/doi.org/10.1016/j.jmro.2023.100145](https://doi.org/10.1016/j.jmro.2023.100145) even though it is not a clear demonstration that this approach is going to work.

8) Giving the costs and the time needed to design and fabricate these MEMS could be useful in particular when we know that high field DNP systems cost many millions.

In summary I believe this is a very fine work and should be published, however I would suggest to help the readers understand better what the authors did, how and why they did it and what kind of practical problems will be able to address with this innovative high field technology.

Reviewer #3 (Remarks to the Author):

This manuscript describes first DNP experiments performed with a nl polymere sample on a single chip integrating NMR and EPR capability at 300 MHz / 200 GHz frequencies respectively. This work is new and very interesting showing the potential to perform DNP on very small samples without the need of expensive sub-THz EPR equipment. I would recommend publication of this exciting work in Nature Comm. after some minor remarks are considered from the authors:

The work with the single MW microresonator is very impressive, also the mw field strength of about 10 G achieved. With the 4 coupled MW resonators it should be also mentioned what average Bmw is expected in this case and what kind of field distribution.

The strong sample size dependence of the observed DNP enhancements for the BDPA/SEBS sample (Figure 5 e/f) should be discussed more in detail. Not only the overall enhancement drops but also the ratio between Solid Effect and Overhauser effect.

This might be related to the point above.

Dear Reviewers,

We thank you for your positive comments about our work and for your useful suggestions/comments/remarks for the improvement of the quality of our manuscript. Here you can find our point-to-point answers (in **black**) to all reviewer suggestions/comments/remarks (in **blue**), which are linked to the changes in the revised manuscript (in **yellow** in the highlighted version). A supplementary information file is created as suggested by the reviewers.

Best Regards,

Reviewer 1:

This paper reports a new chip that has integrated both RF and microwave circuitry to demonstrate a micro-NMR system for performing DNP-enhanced NMR experiments. The authors have provided an excellent introduction to motivate the study, described the design of both the RF and microwave systems in some detail, and demonstrated the excellent performance of their systems. The enhancement appears to be stronger at lower temperatures, indicating the limited microwave power from the chip. I agree with the authors that such on-chip integration of NMR and ESR circuitry could help reduce the complexity of DNP-NMR and make it available for broader applications. Thus, I support the publication of this work after consideration of the following minor comments.

We thank the reviewer for these positive comments.

Title should reflect the full NMR system, instead just the DNP part. For example, “200 GHz single chip dynamic nuclear polarization micro-NMR system”

We thank the reviewer for the valuable advice. We modified the title as follows: 200 GHz single chip microsystems for dynamic nuclear polarization enhanced NMR spectroscopy.

Please include simulations or measurements of the isolation between the RF coil and the microwave coil, perhaps in the supplementary information. Also LNA gain and noise spectrum plots.

As suggested by the reviewer, we have created a Supplementary Information file. In this file, we have included the additional information requested by the reviewer. The isolation between the NMR microcoil and the ESR microcoil is reported in Supplementary Note 1. LNA gain and noise spectrum are reported in Supplementary Note 2. Additionally, the total receiver gain and noise are reported in Supplementary Note 4 for completeness.

On figure 2, please clarify the use of the V_ESR and I_B.

To clarify this point, the following sentences were added to the main text of the revised manuscript:

“The base of the Colpitts pair can be DC biased by a voltage or a current. The current bias I_B of this node through the bias resistor R_P allows for voltage variation at the center of the L_{ESR} inductor and the V_{ESR} node, and hence for the amplitude detection of the ESR signal”

Additionally, we modified the caption of Figure 2 as follows:

“On the left, the block diagram of the NMR circuit is given. VDD_{RX} is the DC supply voltage of all blocks in the NMR receiver. It is connected to several nets and the connections are not shown in the schematic for simplicity. VDD_{LO} is the supply voltage of the LO amplifier. V_{LO} is the down-converting local oscillator signal. V_{NMR} is the output of the NMR detector. On the right, the detailed schematics of the ESR oscillator are given. l_e is the emitter length, w_e is the emitter width, and m is the multiplicity parameter of the heterojunction bipolar transistors. The inductor values are $L_C \cong 10$ pH, $L_E \cong 7$ pH, $L_{ET} \cong 50$ pH, $L_{ESR} \cong 60$ pH, and $L_P \cong 200$ pH. VDD_{ESR} is the DC supply voltage of the oscillator. I_B is the DC bias current of the oscillator. V_T is the tuning voltage of the oscillator. V_{ESR} is the output of the ESR detector.”

Explain how the LO amplifier (Fig 2b) produces different phases at LO_p and LO_n.

As suggested by Reviewer 2, the details of the circuit are removed from the main manuscript and added to the supplementary information file (Supplementary Note 4). In this file, the schematic drawing is updated (Supplementary Figure 5). The different phases are created with the circuit “LO Amplifier” and its sub-blocks “Inverter” and “Delay cell”.

In addition to these changes, the following sentence is added to the description in Supplementary Note 4:

“After the first stages, the LO_N output is obtained by using four inverters in series whereas the LO_P output is obtained by using three inverters and a delay cell in series.”

Figure 4, what are the red and blue lines? It would be great to include a pulse sequence diagram to clarify how the experiments were performed.

In order to clarify this point, we have modified the caption of Figure 4 as follows:

“In (a) the blue curve is the DNP enhanced NMR spectra when the microwave frequency is $(\omega_{OS} - \omega_{OI}) \cong 201.6$ GHz (i.e., to the left of the ESR signal) and the red curve is the DNP enhanced NMR spectra when the microwave frequency is $(\omega_{OS} + \omega_{OI}) \cong 202.2$ GHz (i.e., to the right of the ESR signal) ...

In (b) the blue curve is the DNP enhanced NMR spectra when the microwave frequency is $(\omega_{OS} - \omega_{OI}) \cong 195.7$ GHz and the red curve is the DNP enhanced NMR spectra when the microwave frequency is $(\omega_{OS} + \omega_{OI}) \cong 196.3$ GHz.”

A pulse sequence diagram is now provided in Supplementary Note 3 and the following sentences have been added to the manuscript to clarify this point:

“All NMR and DNP enhanced NMR experiments reported in this work are conventional free induction decay (FID) measurements performed after a single $\pi/2$ pulse. The microwave

excitation from the integrated ESR oscillator is permanently on during all measurements. The pulse sequence diagram is given in Supplementary Note 3.”

For the DNP experiment, the paragraph starting from line 195, please explain the choice of samples and what is the source of ESR.

We have used BDPA:PS samples because they have been used in several previous DNP experiments. We have used, maybe for the first time, also BDPA:SEBS samples because they allow to achieve a larger SNR due to their narrower line at room temperature. The “source” of the ESR signals observed in these samples are the BDPA molecules dispersed in the polymeric matrix of PS and SEBS.

In order to clarify this point, we have modified the Methods section, sample preparation subsection as follows:

“In order to characterize the performance of the realized single chip DNP microsystems, we prepared polystyrene (PS) and styrene-ethylene-butylene-styrene (SEBS) solid polymers doped with 2% α,γ -bisdiphenylene- β -phenylallyl (BDPA/benzene, 152560, Sigma-Aldrich). The ESR linewidth of BDPA in these polymers is in the order of 10 G (28 MHz), hence it is suited for observing well separated solid effect (SE) lines in the DNP enhancement profile at the ^1H Larmor frequency of 300 MHz, corresponding to a field offset for the solid effect of about ± 110 G. BDPA:PS has been investigated in several previous DNP works [53,54,69,70] whereas BDPA:SEBS has not been investigated yet. At room temperature, the ^1H NMR linewidth of PS is of about 50 kHz whereas the one of SEBS is of about 1 kHz, presumably due to motional narrowing as observed also in other elastomers above their glass temperature [24,29,56]. The narrower linewidth of the SEBS sample enables to obtain a larger signal-to-noise ratio and, hence, to perform faster measurements at room temperature.”

It would be useful to demonstrate the microwave power dependence of the NMR enhancement as a way to estimate the degree of saturation of ESR in the current system.

We thank the reviewer for the valuable suggestion. It would be indeed interesting to investigate the degree of saturation by changing the microwave field. Unfortunately, in the current DNP microsystems, the microwave field amplitude B_{1e} can be changed only by a factor of two. The minimum microwave field is determined by the start-up condition of the oscillator, whereas the maximum microwave field is limited by the maximum supply voltage that we can apply to the oscillator. In the future, a larger variation of the effective microwave field B_{1e} might be obtained by a different oscillator design or by pulse modulation of the microwave frequency from resonance to out-of-resonance conditions with variable duty cycle using the oscillator tuning voltage V_T .

To clarify this point, we have added the following paragraph to the Discussion section: *“The microwave field produced by the microcoils of the integrated oscillators can be changed by less than a factor of two by a variation of the oscillator supply voltage VDD_{ESR} and oscillator current I_B . The minimum microwave field is determined by the start-up condition of the oscillator, whereas the maximum microwave field is limited by the maximum supply voltage that we can apply to the oscillator (about 1.5 V). In the future, a larger variation of the effective microwave field might be achievable by a different oscillator design or by pulse modulation with a variable duty cycle of the microwave*

frequency from resonance to out-of-resonance conditions using the oscillator tuning voltage V_T .

Reviewer 2:

The manuscript describes the successful design and construction of two single chip microsystems capable to perform dynamically nuclear polarized magnetic resonance on solids doped with paramagnetic species.

Overall, the manuscript is of very high quality and in my eyes, it is a tour de force in microelectronics combined with advanced magnetic resonance. Achieving this kind of sensitivity a miniaturized integration deserves publication. Given the broad audience of Nature communications I would suggest the authors to revise this document and address some of the following points:

We thank the reviewer for these positive comments.

1) DNP is by no means a generic technique and even though a lot of efforts have been placed to push it the need for unpaired electrons in the sample reduces its applicability. Is there a benefit going microscopic for this drawback of DNP?

We agree that the need to “dope” the samples with unpaired electrons reduces the applicability of the DNP in both microscopic and macroscopic samples. However, we can comment on the particular importance of using DNP for NMR on microscopic samples.

In order to clarify this point, we added the following sentences in the Introduction section of the main text:

“The main difficulty in performing NMR experiments in microscopic samples is the poor SNR. In some experiments, increasing the averaging time to improve the SNR is not possible, such as in the case of biological samples which might evolve/deteriorate over the required experimental time to achieve a sufficiently larger SNR. In these cases, even a modest DNP enhancement would be of crucial importance for the feasibility of the experiment.”

2) The motivations of the authors to develop these high field microsystems is claimed to be: 1) better resolution, 2) higher thermal polarization. In my eyes both arguments are not valid: 1) DNP experiments in higher and higher magnetic fields have until now given more problems than solved (broadening due to disorder increases thus resolution is heavily impacted in poorly crystalized solids and amorphous glasses, heating becomes an issue, more expensive and specialized hardware, lower enhancements, etc.). 2) Given that the polarization comes from the electrons increasing the field has much less effect (linear scaling) compared to reducing the temperature (nonlinear). Therefore, the main valid arguments I see is cost and the possibility to measure many miniature samples (maybe even in parallel with arrays?). The authors should rework a bit these points and make a more detailed motivation list with pros and cons in their final manuscript version.

In general, we agree with the reviewer that the efficient use of DNP at high fields is still problematic in several aspects. But as mentioned in the manuscript and remarked also by the reviewer, this is also a strong motivation for our low cost single chip approach.

The low cost single chip approach might allow a larger community of researchers to study the DNP at high field and to find suitable “receipts” to make it more effective.

We agree with the reviewer that increasing the magnetic field does not necessarily allow to achieve a better “resolution”. As the magnetic field increases, the frequency separation between peaks due to the chemical shift is increased and hence might result in more resolved peaks, but only if the broadening caused by “susceptibility mismatches” and other field dependent effects increases less.

The non-enhanced NMR signal detected inductively is proportional to the thermal nuclear magnetization and the frequency of operation. Both the thermal nuclear magnetization and the frequency of operation increase linearly with the magnetic field. As a result, the non-enhanced NMR signal depends, in first approximation, quadratically on the magnetic field. Since the noise is, again in first approximation, independent of the frequency and field, the signal-to-noise ratio of non-enhanced NMR improves also quadratically with the magnetic field. The DNP enhanced NMR signal depends additionally on the enhancement factor, which is dependent on the magnetic field, the temperature, the specific sample (solid, liquid, concentration and specific “unpaired electron” dopant, etc.), and other experimental conditions. In several samples, the DNP enhancement is reduced at higher magnetic fields. However, several recent works show that the DNP enhancement is reduced less than previously experimentally observed and theoretically predicted. Depending on the specific sample and doping agent, there is most probably an optimum magnetic field at which the enhanced NMR signal and finally the SNR is maximum. This optimum field is most probably significantly larger than 1 T but might be indeed significantly lower than 28 T (the maximum field of current NMR spectrometers). As mentioned above, the single chip approach proposed in this work is well suited tool also for “fundamental” studies to obtain high DNP enhancements at high fields. Thanks to its dramatically lower cost with respect to conventional solutions, these studies are potentially accessible to a larger community of researchers.

As suggested by the reviewer we have changed the introduction section to explain the motivations towards higher field DNP and emphasize better the stronger parts of our work. We have adjusted the second paragraph of the introduction section with the advantages of single chip approach as also mentioned in the previous point of the reviewer and the third paragraph with the explanation of the higher field approach as follows:

“DNP enhanced NMR experiments are performed in a wide range of magnetic fields, from zero to more than 20 T [1,11,41-44]. Operation at higher magnetic fields allows for larger chemical shifts and potentially larger SNR. The larger chemical shifts improve the spectral information content if field induced broadening effects are not dominant. The overall quadratic dependence on the magnetic field of the thermal nuclear magnetization and the frequency of operation might largely compensate for the enhancement reduction generally observed at higher fields and, hence, allow to achieve a larger SNR. As a consequence, the previous two advantages make the exploration of DNP methods at high fields interesting and potentially impactful.”

Additionally, we have modified the last paragraph of the introduction with additional text to emphasize the stronger parts of our work as follows:

“Due to these strong motivations to operate at higher static magnetic fields, in this work we designed and characterized single chip integrated DNP microsystems operating at 200 GHz (ESR)/300 MHz (NMR). The realized single chip microsystems allow to perform continuous wave ESR experiments by field or frequency sweeps and DNP enhanced NMR experiments. Conventional high field DNP methodologies require the use of relatively expensive and bulky microwave/quasi-optical sources and waveguides [43,45-47]. The integration of the microwave source and ESR detector together with the NMR sensor allows to study DNP enhanced NMR without the need for external microwave components. The microwave generation and the microwave detection are performed in situ by the single chip microsystems integrated on an area of less than 1 mm². As a result, these single chip integrated ESR, NMR, and DNP-NMR detectors are suitable for the miniaturization of the probe, for the reduction of the losses and complexity of the connections, and for the realization of dense arrays of detectors for parallel spectroscopy of several samples in the same magnet. By enabling simultaneous measurements, arrays of such sensors reduce the time and the cost of sample characterization. Additionally, the single chip approach might allow for a better SNR for volume limited samples in the nanoliter and subnanoliter range tightly matched to the sensitive volume of the detector with respect to conventional bulky inductive probes optimized for microliter and larger sample volumes. By suppressing the need for external microwave sources and microwave connections, the single chip approach proposed here reduces drastically the cost and the complexity of the DNP instrumentation and, hence, should allow for more widespread use and study of DNP methodologies, particularly for nanoliter and subnanoliter samples.”

3) The document is highly technical and for the general reader hard to follow. Maybe some technical details that are not necessary for following the flow of the work could be added in Supplementary material? Maybe in these SI they could also add more detailed drawings in case someone wished to reproduce this work (there were no financial interests disclosed).

In the revised manuscript we have implemented several changes to follow this suggestion from the reviewer. The schematics in Figure 2 are simplified and the details of the NMR circuitry are moved to Supplementary Note 4 with additional information. The schematics of the current source in the OPAMP, delay cell and inverters have also been added to Supplementary Fig. 5. As a result, this figure contains the drawings of all the circuit blocks of the NMR receiver down to the single transistor level. Similarly, Figure 2 contains the ESR circuitry down to the single transistor level.

4) Figure 2: l_e , w_e and m are not defined as far as I can see and have nm units.

We thank the reviewer for pointing out the missing definitions. In the revised manuscript these definitions are added to the caption of Figure 2 as follows:

“ l_e is the emitter length, w_e is the emitter width, and m is the multiplicity parameter of the heterojunction bipolar transistors.”

5) Some parts of the text are repetitive (page 5 for example, top paragraph).

We thank the reviewer for the remark. The pointed part is about the definitions of the two NMR microcoils for the single oscillator and the oscillator array microsystems. To reduce the repetition, we rephrase it as follows:

“The NMR receiver electronics are the same as the one of the single oscillator DNP microsystem, except for the NMR microcoil which is slightly larger. It has an external diameter of 250 μm (50 μm larger compared to the NMR microcoil of the single oscillator microsystem) and the same wire width, thickness, spacing, and number of turns.”

We removed a few other small repetitions through the text in the final manuscript.

6) Background from protons seems to be a big issue, but no sophisticated background suppression techniques from NMR were developed/applied. The NMR user is going to wonder if these are possible with these MEMS or not and why. The authors should explain a bit more in detail.

We thank the reviewer for this interesting remark. At the moment we have not found a background suppression technique that could be efficiently applied to our situation (static conditions, ^1H NMR, background signal having similar linewidth and chemical shift with respect to the sample under investigation, low SNR due to small sample volumes). It is important to remark that the background NMR signal is caused by the high concentration of protons in the printed circuit board (PCB) used in this work. The single chip microsystem is made of materials that do not contain protons. Hence, it should be possible to solve the proton background problem by using a PCB made of proton-free materials, which are not standard but are commercially available. To clarify this point, in the revised manuscript we changed the discussion as follows:

The experiments performed on solid samples having short T_2 suffered from the presence of a large NMR background signal having a similar linewidth. The background signal is generated by the protons contained in the PCB supporting the single chip integrated microsystem. The single chip integrated microsystem is made of materials which do not contain protons. To drastically reduce the background signal, hence allowing for a more accurate determination of the DNP enhancement, PCBs that do not contain ^1H nuclei, such as those based on alumina or polytetrafluoroethylene (PTFE), could be used. Another possible solution to the background signal problem is the co-integration on a single chip of the NMR excitation coil, the NMR transmitter, and the NMR receiver, as previously reported in a single chip 10 GHz DNP microsystem [47]. The localized NMR excitation produced by integrated microcoil drastically reduces the contribution to the measured signal from the protons contained in the PCB.

7) The fact that the samples are static and not spinning at the magic angle limits tremendously the applicability of these devices towards conventional solid-state NMR studies. The authors mention inductive coupling under MAS as a solution without any references to working examples/experiments. Similarly the reference to the optical MAS is cryptic, hard to find and could be possibly updated to [/doi.org/10.1016/j.jmro.2023.100145](https://doi.org/10.1016/j.jmro.2023.100145) even though it is not a clear demonstration that this approach is going to work.

We thank the reviewer for pointing out the missing reference and the updated reference. In the revised manuscript we have introduced the reference to the inductive coupling under MAS ([/doi.org/10.1038/nature05897](https://doi.org/10.1038/nature05897)) and updated the reference to the optical MAS ([/doi.org/10.1016/j.jmro.2023.100145](https://doi.org/10.1016/j.jmro.2023.100145)).

To clarify the case of the inductive coupling under MAS conditions in the revised manuscript we have added the following sentence:

“A similar approach has been previously demonstrated, although in the simpler case of a spinning microcoil inductively coupled with a static coil connected to the probe electronics [62].”

To clarify the case of optical MAS in the revised manuscript we have modified the following sentence: *“Optical MAS might allow to exceed the spinning frequency limit of the current MAS techniques by an order of magnitude. Preliminary experiments in this direction are reported in Ref. [63] where the experiments on the optical levitation and rotation of subnanoliter samples are reported but the possibility to perform NMR/ESR/DNP experiments on such samples in non-static conditions is still to be demonstrated”*.

8) Giving the costs and the time needed to design and fabricate these MEMS could be useful in particular when we know that high field DNP systems cost many millions.

We agree with the reviewer that this information is indeed interesting, especially for a reader with no experience in integrated circuit design.

To follow this suggestion from the reviewer, we added the following information in Supplementary Note 6:

“The two integrated circuits are realized using a 130 nm SiGe technology (IHP SG13G2Cu). This technology is accessible throughout Europractice (<https://europractice.com/technologies/asics/ihp/>). The price for the fabrication of 40 identical chips each having an area of about 1 mm² is about 5000 Euros. This is the price for a production of 40 chips. The mass production of such chips would reduce the cost down to the 1 Euro/chip level. The time needed for the design of the chips reported in this work was about three months for one full-time-equivalent Ph.D. student with previous experience in integrated circuit design. The time interval between the submission of the design to receiving of the fabricated chips was about six months.”

In summary I believe this is a very fine work and should be published, however I would suggest to help the readers understand better what the authors did, how and why they did it and what kind of practical problems will be able to address with this innovative high field technology.

We thank the reviewer for the positive comments. We believe that thanks to the remarks and suggestions of the reviewer, the revised manuscript is clearer and contains additional useful information (in the main text and the added supplementary information file).

Reviewer 3:

This manuscript describes first DNP experiments performed with a nl polymere sample on a single chip integrating NMR and EPR capability at 300 MHz / 200 GHz frequencies respectively. This work is new and very interesting showing the potential to perform DNP on very small samples without the need of expensive sub-THz EPR equipment. I would recommend publication of this exciting work in Nature Comm. after some minor remarks are considered from the authors:

We thank the reviewer for these positive comments.

The work with the single MW microresonator is very impressive, also the mw field strength of about 10 G achieved. With the 4 coupled MW resonators it should be also mentioned what average Bmw is expected in this case and what kind of field distribution.

To clarify this point, an explanation and a figure (Supplementary Figure 5) are added in the supplementary information file (Supplementary Note 5). Additionally, the following sentence is added to the Method section of the main text:

“The microwave magnetic field produced by the array of four oscillators is estimated by a combination of ADS and Cadence simulations as for the one produced by the single oscillator. This simulation shows that the microwave current running in each of the four microcoils is almost identical to the one running in the single oscillator microcoil. Since the four oscillators are frequency and phase locked, the four microcoils behave as one larger microcoil (see Supplementary Note 5). The diameter of this larger microcoil is approximately two times larger than the one of the single microcoil. Hence, the microwave magnetic field produced by the array of four oscillators is approximately a factor two smaller compared to the one produced by the single oscillator.”

The strong sample size dependence of the observed DNP enhancements for the BDPA/SEBS sample (Figure 5 e/f) should be discussed more in detail. Not only the overall enhancement drops but also the ratio between Solid Effect and Overhauser effect. This might be related to the point above.

Two phenomena might contribute to the observed change in the ratio between the solid effect and the Overhauser effect in the two experiments performed with different repetition time T_r and different sample volumes: 1) the different build-up time (longer for the solid effect) and 2) the different saturation level at a given microwave field (higher for the Overhauser effect).

In order to clarify this point, we have introduced additional measurement results in Fig. 5 of the main text ((c) and (g)). This measurement is performed with the same small sample as in (f) but with a repetition time of 0.1 s instead of 1 s. Additionally, we commented on the measurement results according to the repetition time and sample size in the main text as follows:

“The DNP enhancement curves shown in Fig.5 (f), (g) and (h) are measured with the same sample, while the sample size of Fig.5 (h) is 16 times larger. The oscillator is biased at the same values, whereas the repetition time T_r is longer for the curve in Fig. 5 (f). The Overhauser enhancements are similar for both DNP curves in Fig.5 (f) and (g) (about 1.2) whereas the solid effect enhancement is larger for Fig.5 (f) compared to the Fig.5 (g) (about 1.7 and 1.45, respectively). This is most probably due to the longer build up time of the solid effect compared to the Overhauser effect. The Overhauser enhancements are also similar for both DNP curves Fig.5 (g) and (h) whereas the solid effect enhancement is larger for Fig.5 (g) compared to Fig.5 (h) (about 1.45 and 1.05, respectively). This behavior is probably due to the Overhauser effect being easier to saturate with respect to the solid effect [53]. Since for the larger sample, the average microwave magnetic field is weaker, we would expect that the ratio between the Overhauser and the solid effect is increased in the larger sample, which is consistent with our experiments.”

REVIEWERS' COMMENTS

Reviewer #1 (Remarks to the Author):

The authors have addressed all my questions and I recommend its publication as is.

Reviewer #2 (Remarks to the Author):

Thank you for addressing the points that I made. the manuscript now reads much better. Congratulations on your achievement.

Reviewer #3 (Remarks to the Author):

The authors of this manuscript answered all the questions and remarks of all three reviewers of this article. I suggest now publication of this very nice and comprehensive work in Nature Comm.